# Pre-activation Distributions Expose Backdoor Neurons

**Runkai Zheng**[1]*, **Rongjun Tang**[2]*, **Jianze Li**[3], **Li Liu**[3]†
[1]School of Data Science,
The Chinese University of Hong Kong, Shenzhen
[2]School of Science and Engineering,
The Chinese University of Hong Kong, Shenzhen
[3]Shenzhen Research Institute of Big Data,
The Chinese University of Hong Kong, Shenzhen
`rkteddy@outlook.com, rongjuntang@link.cuhk.edu.cn`
`lijianze@gmail.com, liuli@cuhk.edu.cn`

## Abstract

Convolutional neural networks (CNN) can be manipulated to perform specific behaviors when encountering a particular trigger pattern without affecting the performance on normal samples, which is referred to as backdoor attack. The backdoor attack is usually achieved by injecting a small proportion of poisoned samples into the training set, through which the victim trains a model embedded with the designated backdoor. In this work, we demonstrate that backdoor neurons are exposed by their pre-activation distributions, where populations from benign data and poisoned data show significantly different moments. This property is shown to be attack-invariant and allows us to efficiently locate backdoor neurons. On this basis, we make several proper assumptions on the neuron activation distributions, and propose two backdoor neuron detection strategies based on (1) the differential entropy of the neurons, and (2) the Kullback-Leibler divergence between the benign sample distribution and a poisoned statistics based hypothetical distribution. Experimental results show that our proposed defense strategies are both efficient and effective against various backdoor attacks. Source code is available here.

## 1 Introduction

*Convolutional neural networks* (CNNs) have achieved tremendous success during the past few years in a wide range of areas. However, training a CNN from scratch involves a large amount of data and expensive computational costs, which is sometimes infeasible. A more practical strategy is to obtain pretrained models or utilize public datasets from a third party, which brings convenience but also raises severe security problems into the deployment of models. For example, a malicious third party may provide pretrained models embedded with a designated backdoor, such that the model will have a predefined response to some specific pattern, which is also called the *trigger*. More realistically, the attacker can inject only a small proportion of malicious data into the public dataset to mislead the trained model, which is referred to as *backdoor poisoning attacks* [24]. For instance, the malicious data can be created by patching a particular pattern into the benign data and changing the label to the desired target. The correlation between the trigger and the specified target label will be learned by the models during the training time. In this way, the infected model will misclassify the input to the attack target when the pattern is patched, while behaving normally otherwise, as shown in Figure 1.

According to previous studies, it was empirically found that an infected model always possesses one or more neurons that have high correlation with the trigger activation, and pruning these neurons can

---

*Eual Contribution
†Corresponding Author

36th Conference on Neural Information Processing Systems (NeurIPS 2022).

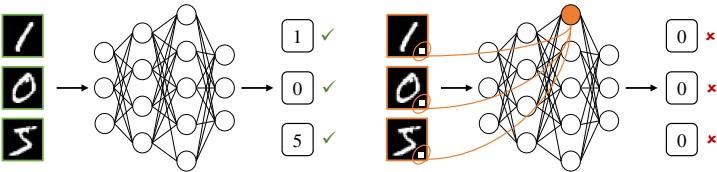

Figure 1: An overview of an infected learning system. The images with a white square are classified as class 0. It is an empirical observation that the backdoor behaviors are always triggered by one or more backdoor neurons.

significantly alleviate the backdoor behaviors, while retaining the model performance [41, 27, 7]. Nevertheless, how to precisely find out these backdoor neurons in an infected model is still a challenging problem, and has attracted a lot of attentions from the community.

In this work, we take an inspection on the pre-activation distributions of infected models on each layer. In general, the pre-activations in each neuron follow an unimodal distribution that can be approximated by a Gaussian distribution. We demonstrate that backdoor neurons do not hold such property. Instead, in a typical backdoor neuron, the pre-activation distributions of benign data and poisoned data present significant different moments, and can be approximated by a mixture of two Gaussian distributions. This property allows us to locate potential backdoor neurons through simple statistical analysis on pre-activations. Specifically, in case the defender has access to the poisoned dataset, the abnormal pre-activation distributions can be directly observed by forward propagating the data. The mixture of benign and poisoned data, where the small proportion of poisoned points being away from the benign mean, leads to a skewed, and even a bimodal distribution. Based on the maximum entropy property of the Gaussian distribution, the skewness will cause a reduction on the differential entropy, compared with a single Gaussian distribution with the same variance. Hence, after standardizing the pre-activation distribution to be unit variance, the abnormal distributions should have lower differential entropy. Those neurons could potentially separate benign and poisoned data. As for another defense setting, in which only an infected model and a set of benign data are provided, we are not able to observe the bimodal distributions since the poisoned data is not available. In this case, we propose to rely on the recorded statistics in *Batch Normalization* (BN) layers. Specifically, if the infected model is trained on poisoned data, the population statistics in backdoor neurons recorded in the BN layer will be different from those of only benign data. More importantly, benign neurons will not exhibit such mismatch in statistics, allowing for differentiation between benign and backdoor neurons. Based on the differential entropy and the statistics discrepancy, we are able to locate and prune potential backdoor neurons to recover the model flexibly under two defense settings.

In summary, our contributions include:

1. We take a deep inspection on the infected model, and summarize the law of pre-activation distributions on poisoned dataset. We find that (1) the standardized entropy of backdoor neurons can be significantly lower than benign neurons, and (2) the BN statistics in infected model are mismatched with the benign sample statistics.

2. We propose to prune potential backdoor neurons based on either the differential entropy of pre-activation distribution or the statistics discrepancy, depending on the defense settings. Under certain assumptions, we claim that both the proposed indices can successfully separate the benign neurons and backdoor neurons by an appropriate threshold.

3. We conduct extensive experiments to verify our assumptions and evaluate our proposed methods, and achieve the state-of-the-art results under two different defense settings.

## 2   Related work

In this section, we briefly discuss recent works in backdoor attack and defense, and a specific branch of related studies on distributional properties of poisoned features.

## 2.1 Backdoor attacks

The most famous *backdoor attack* is introduced in [14], where the adversary injects a small set of targeted label-flipped data with a specific trigger into the training set, leading to a misclassification when predicting the samples with such trigger. To make the trigger pattern even more invisible to human beings, the blending strategy is used in [6] to generate poison images, while the form of natural reflection is utilized in trigger design in [30]. The input image is perturbed in [39] to keep its content consistent with the target label such that the model better memorizes the trigger pattern, and keep it imperceptible to human beings. Moreover, the multi-target and multi-trigger attacks are proposed in [42, 32], and make the attack more flexible and covert. Recently, some sample-specific trigger design strategies [26] are proposed, making the defense against such backdoor attack much harder. Generally, the above attacks can be referred to as the *poisoning based backdoor attacks*.

Under some settings, the attackers can control the training process to inject the backdoor without modifying the training data, referred as the *non-poisoning based backdoor attacks*. This is achieved in [29, 35, 5] through targeted modification of the neurons' weight in a network. Such attacks will not be evaluated in our work due to its strong attack setting.

## 2.2 Backdoor defenses

**Training stage defense.** Under such setting, the defender has access to the training process, so that they can detect and filter the poisoned data or add some restrictions to suppress the backdoor effect in training. Since the poisoned data can be regarded as outliers, different strategies are applied in [10, 12, 1, 38, 15], such as the *robust statistics* in feature space and *input perturbation techniques* to filter them out of training data. Other methods aim at suppressing the backdoor effect during training phase with strong data augmentation methods [2] [25] [34] [31] including *CutMix* [43], *Flip*, *ShrinkPad* [25], *CutOut* and *MaxUp* [13], or *differential privacy* constraints [11, 18].

**Model post-processing defense.** Under some specific scenarios, the defenders are only given a suspicious DNN model without access to the training process or the full training set. Therefore, they must eliminate the backdoor threat with limited resources, such as a small set of clean data. A straightforward way is to reconstruct the trigger, and then mitigate the model with the knowledge of the reversed trigger [40]. Some try to find the relationship between backdoor behaviors and the neurons in a DNN model. Different levels of stimulation to a neuron are introduced in [28] to see how to determine the output activation change, if the model is attacked. Simple neuron pruning strategies are applied in [7] to repair the model, while redundant neuron pruning and fine-tuning are combined in [27] to erase the backdoor effect. Adversarial perturbations are added to the neurons in [41] and precisely prunes the easily-perturbed neurons with more limited clean data requirement and better performance. There are other fine-tuning based methods with the implementation of knowledge distillation [23, 17]. Mode connectivity repair technique [44] is also explored to mitigate the backdoored model. Recently, the *K-Arm optimization* [37] is applied in backdoor detection, helping curtail the threat of backdoor attack.

## 2.3 Distributional properties in poisoned dataset

One branch of research on backdoor learning focuses specifically on using statistical differences in the distribution of benign and poisoned samples to filter out malicious data. Activation clustering [4] uses K-means to separate benign and poisoned samples in feature space. Spectral signatures [38] reveal that feature vectors tend to leave strong signals in the top eigenvectors of their covariance matrix. SPECTRE (Spectral Poison Excision Through Robust Estimation)[15] utilizes tools from robust statistics to amplify the spectral signature by outlier-robust data whitening. Our work differentiated from the above works from the following three aspects: 1) the above works focus on the penultimate layer feature representation, while our work inspects deep into each layer 2) the above works take the representation space from all neurons as a whole, while our finding indicates that the distributional difference only exists in some neurons 3) the above works aim at filtering out poisoned samples for retraining, while our methods directly repair the trained neural network by pruning potential backdoor neurons.

# 3 Preliminaries

## 3.1 Notations

Consider a multi-class classification problem with $C$ classes. Let the original training set $\mathcal{D} = \{(\boldsymbol{x}_i, y_i)\}_{i=1}^N$ contains $N$ i.i.d. sample images $\boldsymbol{x}_i \in \mathbb{R}^{d_c \times d_h \times d_w}$ and the corresponding labels $y_i \in \{1, 2, ..., C\}$ drawn from $\mathcal{X} \times \mathcal{Y}$. Here, we denote by $d_c$, $d_h$ and $d_w$ the number of channels, the height and the width of images, respectively. In particular, we have $d_c = 3$ for RGB images.

As in Section 2.1, the backdoor poisoning attack involves changes to the input images and the corresponding labels on a subset $\mathcal{D}_p \subseteq \mathcal{D}$. In this work, we define the ratio $\rho = \frac{|\mathcal{D}_p|}{|\mathcal{D}|}$ as the *poisoning rate*. We denote the poisoning function to the input images as $\delta(\boldsymbol{x})$. A dataset $\mathcal{D}$ is said to be $\rho$-poisoned if the poisoning rate of the dataset is $\rho$.

Consider a neural network $F(x; \theta)$ with $L$ layers. Denote

$$F^{(l)} = f^{(l)} \circ \phi \circ f^{(l-1)} \circ \phi \circ \cdots \circ \phi \circ f^{(1)},$$

for $1 \leq l \leq L$, where $f^{(l)}$ is a linear function (*e.g.*, convolution) in the $l$-th layer, and $\phi$ is a nonlinear activation function applied element wise. In this work, we may denote $F(x; \theta)$ as $F(x)$ or $F$ for simplicity.

We denote by $\mathcal{W}^{(l)} \in \mathbb{R}^{d_{c'} \times d_c \times d_h \times d_w}$ the weight tensor of a convolutional layer. To do pruning, we apply a mask $\mathcal{M}^{(l)} \in \{0, 1\}^{d_{c'} \times d_c \times d_h \times d_w}$ starting with $\mathcal{M}^{(l)} = \mathbb{1}_{d_{c'} \times d_c \times d_h \times d_w}$ in each layer. Pruning neurons on the network refers to getting a collection of indices $\mathcal{I} = \{(l, k)_i\}_{i=1}^I$ and setting $\mathcal{M}_k^{(l)} = \mathbf{0}_{d_c \times d_h \times d_w}$ if $(l, k) \in \mathcal{I}$. The pruned network $F_{-\mathcal{I}}$ has the same architecture as $F$ but with all the weight matrices of convolutional layers set to $\mathcal{W}^{(l)} \odot \mathcal{M}^{(l)}$, where $\odot$ denotes the Hadamard product.

## 3.2 Differential entropy

To measure the uncertainty of a discrete random variable $Z$, the *entropy* [36, 8] was defined as $H(Z) = -\sum_{z \in Z} p(z) \log p(z)$. At the same time, as an extension of entropy, the *differential entropy* was also introduced for a continuous random variable. More concretely, if $Z$ is a continuous random variable, then it was defined as

$$h(Z) = -\int_Z p(z) \log p(z) dz. \tag{1}$$

An important fact about the differential entropy is that, among all the real-valued distributions supported on $(-\infty, \infty)$ with a specified finite variance, the Gaussian distribution maximizes the differential entropy [8]. In this work, the differential entropy (1) will be utilized to identify the distributions that are far different from a Gaussian distribution.

## 3.3 Backdoor neurons

It was found that there exist one or more neurons that contribute the most to the backdoor behaviors in a infected model [41, 27]. If some of or all of these neurons are pruned, the *attack success rate* will be reduced greatly [41]. In this work, to better quantify the importance of neurons to backdoor behaviors, we would like to introduce the sensitivity of neurons to the backdoor. We first introduce the definition of *backdoor loss* on a specific poisoning function:

**Definition 1.** Given a model $F$ and a poisoning function $\delta$, the backdoor loss on a dataset $\mathcal{D}$ is defined as:
$$\mathcal{L}_{\text{bd}}(f) = \mathbb{E}_{(\boldsymbol{x}, y) \sim \mathcal{D}}[\text{D}_{\text{CE}}(y, f(\delta(\boldsymbol{x})))],$$
where $\text{D}_{\text{CE}}$ denotes the cross entropy loss.

Then:

**Definition 2.** Given a model $F$, the index of a neuron $(l, k)$ and the backdoor loss $\mathcal{L}_{\text{bd}}$, the sensitivity of that neuron to the backdoor is defined as:
$$\alpha(F, l, k) = \mathcal{L}_{\text{bd}}(F) - \mathcal{L}_{\text{bd}}(F_{-\{(l,k)\}}), \tag{2}$$
where $F_{-\{(l,k)\}}$ is the network after pruning the $k$-th neuron of the $l$-th layer.

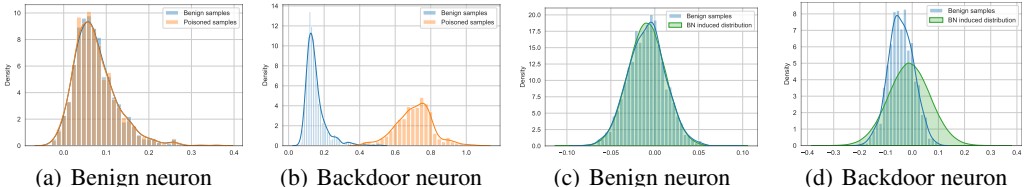

| (a) Benign neuron | (b) Backdoor neuron | (c) Benign neuron | (d) Backdoor neuron |

Figure 2: In (a) and (b), we compare the pre-activation distributions in backdoor neurons and benign neurons. In benign neurons, the pre-activation distributions on benign data and poisoned data are nearly the same, while in backdoor neurons, they show a great difference. In (c) and (d), we plot the empirical distributions with benign samples (in blue) and the BN statistics induced Gaussians (in green). In backdoor neurons, the discrepancy between the empirical and BN-induced distribution is large (all the neurons are selected from infected ResNet-18 trained on CIFAR-10, and use 1,000 images with (poisoned data) or without trigger (benign data) as the inputs).

The backdoor loss is high when the model is infected, and will be reduced when the backdoor effect is alleviated.

Using the quantity defined in (2), we are now able to find the neurons that are mostly correlated with the backdoor behaviors:

**Definition 3.** Given a model $F$ and a threshold $\tau > 0$, the set of backdoor neurons are defined as:

$$\mathcal{B}_{F,\tau} = \{(l,k) : \alpha(F, l, k) > \tau\}. \tag{3}$$

### 3.4 Pre-activation distribution

During the forward propagation of an input $\boldsymbol{x}$, we denote $\boldsymbol{x}^{(l)} = F^{(l)}(\boldsymbol{x}) \in \mathbb{R}^{d_c^{(l)} \times d_h^{(l)} \times d_w^{(l)}}$ as the output of the $l$-th layer. For the $k$-th neuron of the $l$-th layer, the *pre-activation* $\boldsymbol{\phi}_k^{(l)} = \phi(\boldsymbol{x}_k^{(l)})$ is defined as the maximum value of the $k$-th slice matrix of dimension $d_h^{(l)} \times d_w^{(l)}$ in $\boldsymbol{x}^{(l)}$. The reason we choose *pre-activations* instead of *activations* is that the distribution of activations after non-linear function might be distorted. For example, ReLU (rectified linear unit) will cut out all negative values.

It is a common assumption that, for every neuron, the pre-activations before the non-linear function follow a Gaussian distribution, if the network is randomly initialized and the number of neurons is large enough [20]. This assumption is based on the central limit theorem under weak dependence [3]. In a trained network, although this assumption may not strictly hold, the pre-activation of every neuron can be still regarded as approximately following a Gaussian distribution. However, in this work, for the first time, we observe a bimodal pre-activation distribution in backdoor neurons formed by the benign data and poisoned data. This phenomenon is shown in Figure 2, where a typical backdoor neuron is compared with the benign neurons. It can be seen that, after the model is infected, the pre-activation distributions of benign neurons hardly change when the data is poisoned, while the pre-activation distributions of backdoor neurons become significantly different.

## 4 Methodology

### 4.1 Basic assumptions

We now introduce two preliminarily assumptions on the pre-activation distribution of an infected neural network.

**Assumption 1.** Given an infected model $F$, we have $|\mathcal{B}_{F,\tau}| > 0$ for some threshold $\tau > 0$.

This is a primary assumption that guarantees proper pruning of neurons can correct the network's predictions on poisoned samples to some extent. Hence, it is a prerequisite of good performance of all the pruning-based defense methods.

The next assumption provides a precondition for our methods:

**Assumption 2.** Given a model $F$ infected by a poisoning function $\delta$ with a $\rho$-poisoned dataset $\mathcal{D}$, the pre-activation of sample from $\mathcal{D}$ on each single neuron of $F$ follow a Gaussian mixture distribution,

that is:

$$\phi_k^{(l)} \sim (1 - \rho)\mathcal{N}(\mu_k^{(l)}, \sigma_k^{(l)2}) + \rho\mathcal{N}(\hat{\mu}_k^{(l)}, \hat{\sigma}_k^{(l)2}),$$

with

$$|\mu_k^{(l)} - \hat{\mu}_k^{(l)}| \begin{cases} < \epsilon, & \text{if } (l, k) \notin \mathcal{B}_{F,\tau}, \\ \gg \epsilon, & \text{if } (l, k) \in \mathcal{B}_{F,\tau}, \end{cases}$$

and

$$|\sigma_k^{(l)2} - \hat{\sigma}_k^{(l)2}| < \epsilon, \quad \forall k \notin \mathcal{B}_{F,\tau}$$

where $\epsilon > 0$ is a small enough value, $\mu_k^{(l)}$ and $\sigma_k^{(l)2}$, $\hat{\mu}_k^{(l)}$ and $\hat{\sigma}_k^{(l)2}$ are the mean and variance of $\{\phi(F^{(l)}(\boldsymbol{x})_k) : \boldsymbol{x} \sim \mathcal{X}\}$, $\{\phi(F^{(l)}(\delta(\boldsymbol{x}))_k) : \boldsymbol{x} \sim \mathcal{X}\}$, respectively.

This assumes that the mean value of pre-activation distribution of benign and poisoned samples only significantly differentiated in backdoor neurons. This assumption is made based on empirical observation, and our methods work only when this assumption holds.

## 4.2 Entropy-based pruning (EP)

Based on the given assumptions, standardizing the pre-activation distributions (by subtracting the mean and dividing by the standard deviation) will maximize the differential entropy in benign neurons, which approximately follow a standard Gaussian distribution ($\mathcal{N}(0, 1)$). However, in backdoor neurons, the mixture distributions resulting from differences in the moments of Gaussian components cannot be Gaussian distributions, leading to a smaller differential entropy than that of a standardized Gaussian distribution.

**Corollary 1.** Let $\dot{\phi}_k^{(l)} = \frac{\phi_k^{(l)} - \mu_k^{(l)}}{\sigma_k^{(l)}}$ be the standardized pre-activations, then the following inequality is satisfied:

$$h(\dot{\phi}_k^{(l)}) < h(\dot{\phi}_{k'}^{(l)}) \leq h(Z), \quad \forall k \in \mathcal{B}_{F,\tau}, k' \notin \mathcal{B}_{F,\tau},$$

where $Z \sim \mathcal{N}(0, 1)$ is the standardized Gaussian distribution.

This inequality gives a guarantee that with an appropriately chosen threshold, the backdoor neurons can be well separated with the benign neurons.

## 4.3 BN statistics-based pruning (BNP)

BN layer involves using the statistics of a mini-batch to normalize the data in each layer for each neuron. It is known to be able to smooth the optimization landscape, and has gradually become a common setting of neural networks [19]. During inference, BN uses the fixed statistics obtained by averaging the sample statistics of mini-batches during training time, including the mean and the variance. If the model is trained on a poisoned dataset, BN will record the mean and the variance of the poison-benign mixed data. Note that the mean and variance here are not defined on the pre-activations $\phi_k^{(l)}$, but on $\boldsymbol{x}_k^{(l)}$. Based on the above discussions, we know that the poisoned pre-activations in backdoor neurons follow a different distribution from the benign samples. The recorded statistics during training are actually that of the mixture distribution. Hence, we can expect that the BN statistics of a trained backdoor neural network are biased. If we are able to access a small set of benign data, we can calculate an approximation of the true statistics on benign data. Then we calculate the *Kullback-Leibler* (KL) divergence [9] between the sample distribution and the BN induced distribution as the measurement of the bias. By assuming both of the distributions follow Gaussian distributions, we have a closed-form solution:

$$\mathrm{D}_{\mathrm{KL}}(\mathcal{N}_{\mathrm{sample}}^{(l)}, \mathcal{N}_{\mathrm{BN}}^{(l)})_k = \log \frac{\tilde{\sigma}_k^{(l)}}{\sigma_k^{(l)}} + \frac{\sigma_k^{(l)2} + (\mu_k^{(l)} - \tilde{\mu}_k^{(l)})^2}{2\tilde{\sigma}_k^{(l)2}} - \frac{1}{2},$$

where $\mathcal{N}_{\mathrm{sample}}^{(l)} = \mathcal{N}(\mu_k^{(l)}, \sigma_k^{(l)2})$, $\mathcal{N}_{\mathrm{BN}}^{(l)} = \mathcal{N}(\tilde{\mu}_k^{(l)}, \tilde{\sigma}_k^{(l)2})$, $\mu_k^{(l)}$ and $\sigma_k^{(l)2}$ are the statistics obtained from benign samples, $\tilde{\mu}_k^{(l)2}$ and $\tilde{\sigma}_k^{(l)}$ are the BN statistics. Note that BN statistics is the mixture statistics of benign and poisoned distributions. Thus, we have the following corollary:

**Corollary 2.** According to Assumption 2, when $\epsilon \to 0$, the following inequality is satisfied:

$$\mathrm{D_{KL}}(\mathcal{N}_{\mathrm{sample}}^{(l)}, \mathcal{N}_{\mathrm{BN}}^{(l)})_k > \mathrm{D_{KL}}(\mathcal{N}_{\mathrm{sample}}^{(l)}, \mathcal{N}_{\mathrm{BN}}^{(l)})_{k'} = 0, \quad \forall k \in \mathcal{B}_{F,\tau}, k' \notin \mathcal{B}_{F,\tau},$$

The corollary indicates that backdoor neurons should have larger KL divergences than benign neurons, as illustrated in Figure 2(c).

## 4.4 Overview of the two pruning strategies

In Section 3.4, we reveal the discrepancy between the pre-activation distributions in backdoor neurons and that in the benign neurons. This enables fast detecting the neurons that are more related to the backdoor behaviours. The index we choose to detect the abnormal neurons depends on what kind of data we are able to access.

**Mixture training data** In this case, the victim is given a poisoned training dataset with a specified poisoning rate $\rho$. Our goal is to obtain a benign model based on the poisoned dataset. To achieve this, we first train an infected model $F$ on the poisoned dataset. The resulting model should have a certain number of backdoor neurons based on empirical observation and assumption. Since $\rho > 0$ for the dataset, all the neurons follow Gaussian mixture distributions, and we have $h(\dot{\boldsymbol{x}}_k^{(l)}) < h(\dot{\boldsymbol{x}}_{k'}^{(l)})$ for all $k \in \mathcal{B}_{F,\tau}, k' \notin \mathcal{B}_{F,\tau}$. This implies that with an appropriate threshold $\tau_h^*$, we can perfectly separate the benign neurons and backdoor neurons, which can be formulated as:

$$\exists \tau_h^*, \quad h(\dot{\boldsymbol{x}}_k^{(l)}) < \tau_h^*, \quad \forall k \in \mathcal{B}_{F,\tau},$$
$$h(\dot{\boldsymbol{x}}_{k'}^{(l)}) > \tau_h^*, \quad \forall k' \notin \mathcal{B}_{F,\tau}.$$

Setting the threshold $\tau_h^*$ is crucial to the solution, and it is a trade-off between the accuracy on benign samples and that on the backdoored samples. Note that $|\mathcal{B}_{F,\tau}^{(l)}| << d_c^{(l)}$. We can treat the low entropy neurons as outliers in each layer, and set different thresholds for different layers. Specifically, let $h^{(l)} = [h(x_1^{(l)}), h(x_2^{(l)}), \ldots, h(x_{d_c^{(l)}}^{(l)})]^T \in \mathbb{R}^{d_c^{(l)}}$ be a vector of differential entropy of the $l$-th layer calculated from the poisoned dataset. Then we set $\tau_h^{(l)} = \bar{h}^{(l)} - u_h \cdot s_h^{(l)}$, where $\bar{h}^{(l)} = \frac{1}{d_c^{(l)}} \sum_{k=1}^{d_c^{(l)}} h_k^{(l)}$ and $s_h^{(l)} = \sqrt{\frac{1}{d_c^{(l)}} \sum_{k=1}^{d_c^{(l)}} (h_k^{(l)} - \bar{h}^{(l)})^2}$ are the mean and standard deviation of $h^{(l)}$, $u_h$ is a hyperparameter controlling how low the threshold is. Then we have a set of indices of potential backdoor neurons $\mathcal{I}_h = \{(l,k) : h_k^{(l)} < \tau_h^{(l)}\}$. Finally, we prune the infected model $F$ using $\mathcal{I}_h$, which results in a final model $F_{-\mathcal{I}_h}$.

**Benign training data** This is the case that the victim is given a trained poisoned model $F$ with a small set of benign data. Our goal is to utilize the benign data to clean up the poisoned model and eliminate the backdoor threat. Similar to the pruning process based on differential entropy, we first construct a vector of KL divergences of all neurons for each layer $K^{(l)} = [K_1^{(l)}, K_2^{(l)}, \ldots, K_{d_c^{(l)}}^{(l)}]^T \in \mathbb{R}^{d_c^{(l)}}$ according to equation (4). We set $\tau_K^{(l)} = \bar{K}^{(l)} + u_K \cdot s_K^{(l)}$, where $\bar{K}^{(l)} = \frac{1}{d_c^{(l)}} \sum_{k=1}^{d_c^{(l)}} K_k^{(l)}$ and $s_K^{(l)} = \sqrt{\frac{1}{d_c^{(l)}} \sum_{k=1}^{d_c^{(l)}} (K_k^{(l)} - \bar{K}^{(l)})^2}$ are the mean and standard deviation of $K^{(l)}$, $u_K$ is a hyperparameter. The set of selected neurons is $\mathcal{I}_K = \{(l,k) : K_k^{(l)} > \tau_K^{(l)}\}$ and the pruned model can be represented as $F_{-\mathcal{I}_K}$.

# 5 Experiments

## 5.1 Implementation details

**Datasets** In this section, the experiments are conducted on two influential benchmarks, CIFAR-10 [21] and Tiny-ImageNet [22]. We use $90\%$ of the data set for training, the rest of the data is used for validating or recovering the poisoned model.

**Models**   We use ResNet-18 [16] as the baseline model to evaluate our proposed method, and compare it with other methods. We train the network for 150 epochs on CIFAR-10 and 100 epochs on Tiny-ImageNet with SGD optimizer. The initial learning rate is set to 0.1 and the momentum is set to 0.9. We adopt the cosine learning rate scheduler to adjust the learning rate. The batch size is set to 128 by default.

**Attacks**   Our experiments are based on both the classical and the most advanced attack strategies, including the *BadNet* [14], *Clean Label Attack* (CLA) [39], *Reflection Backdoor* (Refool) [30], *Warping-based poisoned Networks* [33], *Blended backdoor attack* (Blended) [6], *Input-aware back-door attack* (IAB) [32] and *Sample Specific Backdoor Attack* (SSBA) [26]. For BadNets, we test both the *All-to-All* (A2A) attack and *All-to-One* (A2O) attack, *i.e.*, the attack target labels are set to $y_t = (y + 1) \mod C$, or one particular label $y_t = C_t$, respectively. The target for A2O attacks of all the attack strategies is set to class 0. The triggers for BadNets and CLA are set to randomly generated patterns with size 3×3 for CIFAR-10 and 5×5 for Tiny-ImageNet. The poisoning rate is set to 10% by default. Note that, due to the image size restraint, SSBA is only performed on Tiny-Imagenet.

**Defenses**   We conduct experiments under two defense settings, one of which allows the defender to access the poisoned training set, while the other only has a small clean data set. Both the defense goals are to obtain a clean model without backdoor behaviors. We compare our approaches with the $l_\infty$ *pruning* [7], *fine-tuning* (FT), *fine-pruning* (FP) [27], *neural attention distillation* (NAD) [23] and *adversarial neuron pruning* (ANP) [41]. The number of benign samples allowed to access is set to 500 (1%) for CIFAR-10 and 5000 (5%) for Tiny-ImageNet by default. We set the threshold hyperparameter $u_h/u_k$ to 3 on CIFAR-10 and 4 on Tiny-ImageNet of all tested attacks by default.

**Evaluation metrics**   In this work, we use the *clean accuracy* (ACC) and *attack success rate* (ASR) to evaluate the effectiveness of different methods. The ACC for a given model $F$ is defined as:

$$\text{ACC}(F, \mathcal{D}_{\text{test}}) = \sum_{(\boldsymbol{x}, y) \in \mathcal{D}_{\text{test}}} \mathbb{I}\{\arg\max(F(\boldsymbol{x})) = y\},$$

where $\mathbb{I}$ is the *indicator function*. The ASR is defined as:

$$\text{ASR}(F, \mathcal{D}_{\text{test}}) = \sum_{(\boldsymbol{x}, y) \in \mathcal{D}_{\text{test}}, y \neq y_t} \mathbb{I}\{\arg\max(F(\delta(\boldsymbol{x}))) = y_t\},$$

where $y_t$ is the attack target label. The ACC measures the model performance on benign samples, while the ASR reflects the degree of backdoor behavior retainment in the model. Given an infected model, our goal is to reduce the ASR, while keeping the ACC from dropping too much.

### 5.2   Experimental results

**CIFAR-10**   We show the results on CIFAR-10 in Table 1. The recently proposed NAD and ANP perform significantly better than other defense methods, reducing the ASR to a very low level with a slight drop on ACC. However, they also have a significant drop (3 ∼ 4%) on ACC when defending CLA, which is the most robust backdoor attack in our experiments, and ANP even failed when defending BadNets(A2A). Nevertheless, both of our methods successfully eliminate the backdoor (ASR < 1%) with negligible loss on ACC. We even observe a little rise on ACC when defending BadNets by EP. This phenomenon demonstrates that backdoor neurons may hurt the ACC in some way, and thus the ACC will rise when the backdoor neurons are precisely pruned. Overall, our methods achieve the most advanced defense results.

**Tiny-ImageNet**   Tiny-ImageNet is a larger scale dataset with higher resolution images, and it is harder to defend against the attacks performed on it. Note that the A2A attack is absent, since we cannot successfully perform the attack due to the large number of its classes (up to 200). Our experimental results show that all the defense methods suffer from the performance degradation compared with the results in CIFAR-10, and they fail to defend against WaNet with a large ACC drop but even unchanged ASR, especially the ANP and $l_\infty$ defense. This phenomenon shows that the principles for finding backdoor neurons of both ANP and $l_\infty$ don't work in such case. Nevertheless, our methods totally remove the backdoor and the ACC are not even affected, which indicates that our methods can precisely locate the backdoor neurons even on such large scale dataset.

Table 1: Experimental results of the proposed approaches against different attacks compared with other defense methods in CIFAR-10[21].

| Attacks | BadNets (A2O) | | BadNets (A2A) | | CLA | | WaNet | | Blended | | Refool | | IAB | |
|---|---|---|---|---|---|---|---|---|---|---|---|---|---|---|
| | ACC | ASR | ACC | ASR | ACC | ASR | ACC | ASR | ACC | ASR | ACC | ASR | ACC | ASR |
| Origin | 93.86 | 100.00 | 94.60 | 93.89 | 94.99 | 98.83 | 94.11 | 99.67 | 94.17 | 99.62 | 94.24 | 98.40 | 93.87 | 97.91 |
| FT | 92.22 | 2.16 | 92.03 | 60.76 | 92.88 | 95.73 | 92.93 | 9.37 | 93.9 | 90.27 | 91.68 | 17.78 | 91.78 | 9.52 |
| FP | 92.18 | 2.97 | 91.75 | 66.82 | 92.60 | 99.36 | 92.07 | 1.03 | 70.92 | 90.92 | 92.36 | 75.98 | 87.04 | 16.13 |
| $l_\infty$ | 92.12 | 100.00 | 93.67 | 6.67 | 92.75 | 98.76 | 93.48 | 99.74 | 86.99 | 99.77 | 91.19 | 98.47 | 88.37 | 88.48 |
| NAD | 93.36 | 2.43 | 92.18 | 2.06 | 91.36 | 15.31 | 93.08 | 3.05 | 92.72 | 1.61 | 91.64 | 6.74 | 92.11 | 19.45 |
| ANP | 93.47 | 3.53 | 90.29 | 86.22 | 91.13 | 11.76 | **94.12** | **0.51** | 93.66 | 5.03 | 91.71 | 26.96 | **93.52** | 10.61 |
| EP (Ours) | **93.88** | **0.86** | **94.49** | **0.61** | **94.42** | **0.91** | 93.79 | 2.80 | 93.67 | **2.24** | 93.35 | 8.90 | 93.17 | 0.94 |
| BNP (Ours) | 93.60 | 1.60 | 94.25 | 0.72 | 94.14 | 7.03 | 94.05 | 3.39 | **94.17** | 2.71 | **93.69** | **6.48** | 93.15 | **0.64** |

Table 2: Experimental results of the proposed approaches against different attacks compared with other defense methods in Tiny-ImageNet[22].

| Attacks | BadNets (A2O) | | CLA | | WaNet | | Refool | | Blended | | IAB | | SSBA | |
|---|---|---|---|---|---|---|---|---|---|---|---|---|---|---|
| | ACC | ASR | ACC | ASR | ACC | ASR | ACC | ASR | ACC | ASR | ACC | ASR | ACC | ASR |
| Origin | 61.36 | 97.38 | 65.61 | 56.58 | 61.47 | 99.98 | 53.26 | 80.61 | 62.85 | 99.83 | 61.40 | 98.28 | 66.51 | 99.78 |
| FT | 46.93 | 99.84 | 61.19 | 63.20 | 54.28 | 99.96 | 47.09 | 91.77 | 56.83 | 29.12 | 52.39 | 99.10 | 52.39 | 33.19 |
| FP | 35.41 | 99.48 | 62.30 | 39.05 | 53.65 | 100.00 | 42.10 | 86.62 | 59.59 | 99.76 | 52.67 | 98.47 | 53.36 | 31.96 |
| $l_\infty$ | 53.13 | 90.39 | 59.15 | 23.12 | 42.01 | 99.84 | 46.84 | 81.19 | 56.33 | 99.85 | 49.35 | 99.98 | | |
| NAD | 44.20 | 90.13 | 62.80 | 17.35 | 53.40 | 99.98 | 51.06 | 70.63 | 57.35 | 55.60 | 53.32 | 98.85 | 52.52 | 25.08 |
| ANP | 53.85 | 4.02 | 59.69 | 3.64 | 54.82 | 86.98 | 50.67 | **0.21** | **62.49** | **0.61** | **61.39** | 4.67 | 60.98 | 1.01 |
| EP (Ours) | 60.68 | **0.86** | 64.47 | 0.10 | 60.53 | 0.02 | 51.29 | 17.07 | 60.67 | **0.69** | 61.26 | **0.60** | 64.2 | 0.11 |
| BNP (Ours) | **61.60** | 1.60 | **64.86** | **0.05** | **61.58** | **0.01** | **52.41** | 23.79 | 60.77 | 0.85 | **61.30** | **0.60** | **64.64** | **0.01** |

## 5.3 Ablation study

To be fair, we compare BNP with other re-training based methods using 500 benign samples in Section 5.2. However, BNP doesn't require re-training the model, and the samples are just used for detecting the distribution discrepancy. As the statistical differences may be detected with much fewer samples, we now study how the number of samples affects the effectiveness of BNP. We train BadNets, CLA, Refool and Blended on CIFAR-10 with $\rho = 10\%$, and use 10 to 500 benign samples to recover the model using BNP. We record the changes of ACC and ASR with respect to the number of benign samples. The results are shown in Section 5.3. The influence of the number of samples to our methods comes from the randomness on estimating moments. As the number of samples grows, the randomness is reduced and BNP has more stable performance, but the average performances are not improved, except for Refool. Compared with other attacks, Refool clearly needs more samples to reduce the ASR. A possible reason is that the mixture distribution in Refool has closer moments and is harder to distinguish. Besides, we surprisingly find that BNP can recover BadNets, CLA and Blended using only 10 benign samples. We also conduct experiments to show the high correlation between the backdoor neurons and our proposed evaluation metrics, and the results are shown in Appendix D.

## 6 Discussion

The proposed methods are superior to other existing defense methods in the following three aspects:

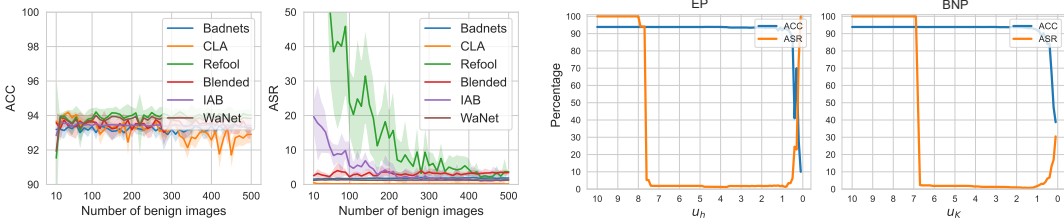

Figure 3: This figure shows the influence of samples used for BNP.

Figure 4: This figure shows the performance of the proposed EP and BNP with different hyperparameter against badnet on CIFAR-10 with ResNet-18.

**Better performance**   As demonstrated in Section 5, both of the proposed methods achieve state-of-the-art results. Moreover, according to the ablation study, the proposed BNP can successfully defend most of the attacks within 10 benign samples, which shows the amazing effectiveness of our proposed methods.

**Higher efficiency**   The proposed methods are highly efficient. We record the running time of several defense methods on 500 CIFAR-10 images with ResNet-18, and show the results in Table 3. It can be seen that both of the proposed methods require less time than the baseline defense methods. Since both methods require scanning on each neuron once, the computational complexity scale linearly with the number of the neurons in the neural network. Therefore, the efficiency of our methods is promised.

Table 3: The overall running time of different defense methods on 500 CIFAR-10 images with ResNet-18.

| Defense Method | FT | FP | NAD | ANP | EP (ours) | BNP (ours) |
|---|---|---|---|---|---|---|
| Runing Time (sec.) | 12.35s | $14.59s$ | $22.08s$ | $25.68s$ | $10.69s$ | **0.39s** |

**More robust to hyperparameter choosing**   One of the most general problems in backdoor defense is the choice of hyperparameters. Under realistic settings, the defenders can only perform defenses without any prior knowledge about poison data, including poisoning rate and examples of poisoned data. So the defenders should carefully tune the hyperparameters, or the ACC and ASR can change suddenly even under small fluctuations of those hyperparameters. In comparison, both of the proposed pruning strategies only require one universal hyperparameter $u$. Moreover, they show reliable consistency against different attacks in the same dataset, only vary from different datasets, which is inevitable. Besides, we leave a wide range of parameters to choose, so that the ACC remains high while the ASR is controlled to a very small number, as shown in Section 5.3.

# 7   Conclusion

In this work, we take an inspection on the characteristics of an infected model, and find backdoor sensitive neurons distinguishable by their pre-activations on poisoned dataset. Specifically, in backdoor neurons, pre-activations from benign data and poisoned data form distribution with extraordinarily different moments. This property makes it possible for defenders to efficiently locate the potential backdoor neurons based on the distributional property of pre-activations. When direct access to the poisoned dataset is available, we propose to measure the mixture property of pre-activations via differential entropy to detect potential backdoor neurons. In another case, where defenders only have access to a benign dataset, we propose to check abnormality of pre-activation distribution based on the inconsistency of the recorded BN statistics and the sample statistics on the given benign dataset. We then do pruning on potential backdoor neurons to recover the model. Experiments show that the proposed defending strategies can efficiently locate the backdoor neurons, and greatly reduce the backdoor threat with negligible loss of clean accuracy. Our approaches achieve superior results compared with all other defense methods under various attacks on the tested datasets. The results shed lights on the field of backdoor defense, and can be a guidance for designing more robust backdoor attacks.

# 8   Acknowledgement

This work is supported by the National Natural Science Foundation of China (No. 62101351), the GuangDong Basic and Applied Basic Research Foundation (No.2020A1515110376), Shenzhen Outstanding Scientific and Technological Innovation Talents PhD Startup Project (No. RCBS20210609104447108), the Key-Area Research and Development Program of Guangdong Province (2020B0101350001), and the Chinese University of Hong Kong (Shenzhen).

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
