# A    Example poisoned images of different attacks

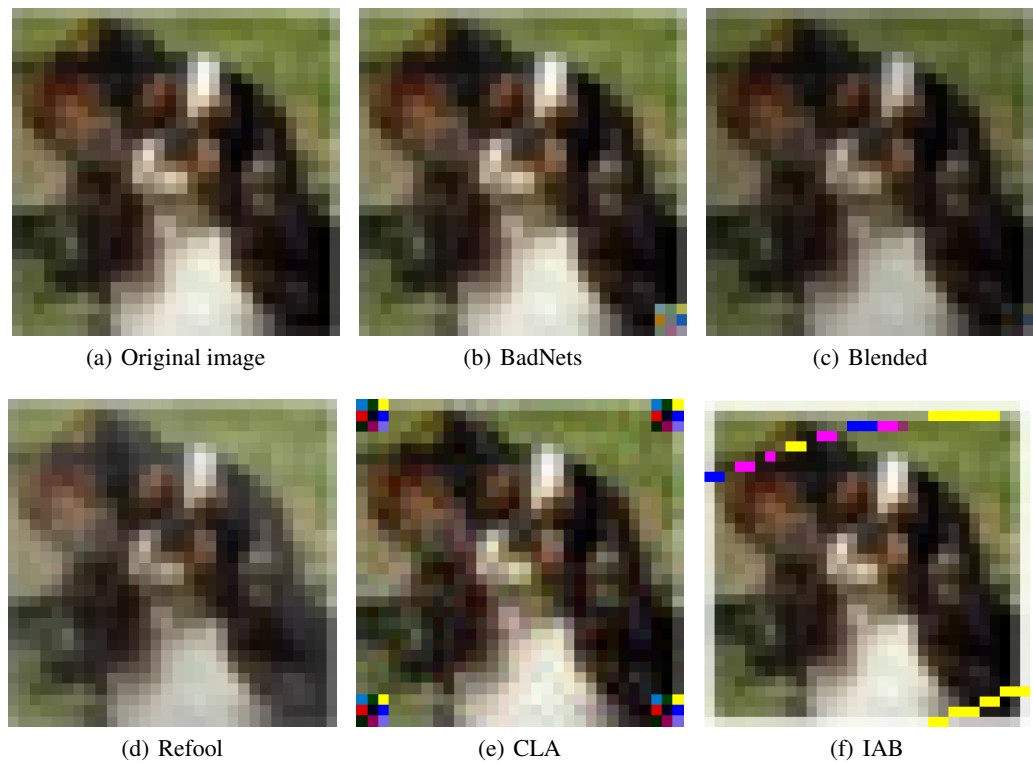

(a) Original image        (b) BadNets        (c) Blended

(d) Refool        (e) CLA        (f) IAB

Figure 5: Examples of the poisoned data on CIFAR-10.

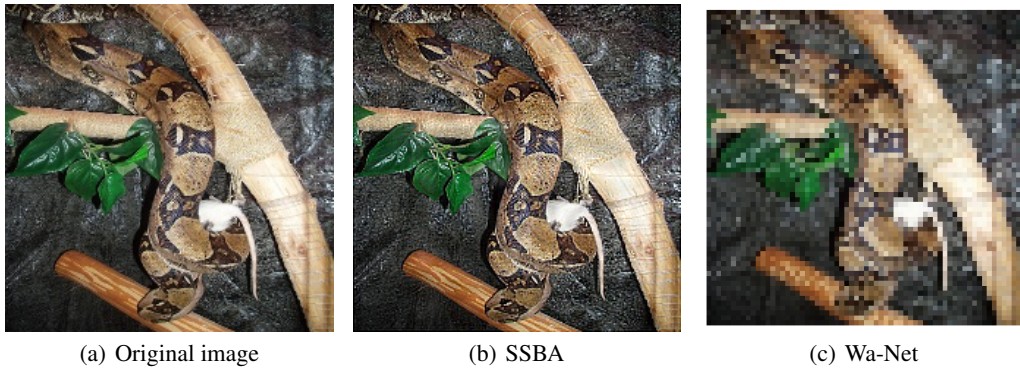

(a) Original image        (b) SSBA        (c) Wa-Net

Figure 6: Examples of the poisoned data on Tiny-ImageNet

# B    Experimental results of different Poisoning Rate

Poisoning rate is crucial to the performance of the proposed methods, since smaller poisoning rate makes the bimodal distributions less distinguishable. In EP, decreasing poisoning rate will increase the entropy of the backdoor neurons, while in BNP, the smaller poisoning rate makes the BN statistics closer to the true benign statistics. Hence, we study the influence of poisoning rate on the performance of the proposed methods. We set the poisoning rate to $1\%$, $5\%$ and $10\%$ for all attacks and show the results in Table 4. Note that CLA fails to attack the model when the poisoning rate is set to $1\%$. In most cases, their performances are acceptable. Nevertheless, there are severe degradation against

CLA and Refool. Specifically, the ASR of CLA and Refool remain over 10% after pruning using EP and BNP. Note that we use default hyperparameter, i.e., $u_h = 3$ and $u_K = 3$ here.

Table 4: Experimental results of the proposed approaches against different attacks compared with other defense methods in CIFAR-10[23].

| $\rho$ | Stage | BadNets (A2O) | | BadNets (A2A) | | CLA | | WaNet | | Blended | | Refool | | IAB | |
|---|---|---|---|---|---|---|---|---|---|---|---|---|---|---|---|
| | | ACC | ASR | ACC | ASR | ACC | ASR | ACC | ASR | ACC | ASR | ACC | ASR | ACC | ASR |
| 1% | Origin | 95.03 | 99.94 | 94.75 | 88.57 | 88.96 | 4.73 | 94.76 | 46.82 | 94.17 | 99.62 | 93.08 | 99.59 | 93.22 | 64.00 |
| | EP | 94.82 | 0.91 | 94.17 | 0.73 | 87.96 | 0.65 | 93.67 | 14.17 | 94.46 | 2.29 | 91.77 | 24.08 | 92.73 | 5.21 |
| | BNP | 92.22 | 2.16 | 93.16 | 7.99 | 88.03 | 0.84 | 94.64 | 1.24 | 92.89 | 2.44 | 90.99 | 21.22 | 93.17 | 4.19 |
| 5% | Origin | 94.29 | 99.99 | 94.26 | 92.78 | 95.53 | 92.23 | 94.00 | 94.55 | 94.53 | 81.33 | 94.35 | 97.98 | 92.70 | 65.50 |
| | EP | 93.83 | 0.83 | 93.67 | 0.70 | 94.43 | 15.91 | 92.73 | 10.13 | 94.44 | 5.49 | 92.75 | 4.51 | 92.29 | 1.83 |
| | BNP | 93.61 | 0.67 | 93.99 | 5.64 | 94.65 | 12.06 | 94.17 | 1.78 | 93.37 | 9.21 | 92.30 | 2.08 | 92.74 | 2.14 |
| 10% | Origin | 93.89 | 100.00 | 94.60 | 93.89 | 94.99 | 98.83 | 94.11 | 99.67 | 94.17 | 99.63 | 94.24 | 98.40 | 93.87 | 97.91 |
| | EP | 93.88 | 0.86 | 94.49 | 0.61 | 94.42 | 0.91 | 93.79 | 2.80 | 93.67 | 2.24 | 93.35 | 8.90 | 93.17 | 0.94 |
| | BNP | 93.60 | 1.60 | 94.25 | 0.72 | 94.14 | 7.03 | 94.05 | 3.39 | 94.17 | 2.71 | 93.69 | 6.48 | 93.15 | 0.64 |

## C   Experimental results on Wide-ResNet

We conduct experiments on WideResNet-28-1, and the results are shown in Table 5.

Table 5: Additional experimental results on another widely-used architecture, WideResNet-28-1.

| Attacks | Backdoored | | EP | | BNP | |
|---|---|---|---|---|---|---|
| | ACC | ASR | ACC | ASR | ACC | ASR |
| BadNets (A2O) | 91.62 | 99.99 | 90.30 | 1.71 | 91.37 | 1.61 |
| BadNets (A2A) | 92.53 | 92.03 | 91.50 | 0.90 | 91.43 | 1.56 |
| CLA | 92.81 | 80.14 | 91.55 | 10.24 | 91.91 | 4.30 |
| Refool | 91.53 | 97.28 | 91.11 | 0.74 | 90.29 | 2.87 |
| WaNet | 91.78 | 91.92 | 92.28 | 0.67 | 91.84 | 0.56 |
| Blended | 91.65 | 99.78 | 91.59 | 6.40 | 91.59 | 1.52 |
| IAB | 90.32 | 88.08 | 90.97 | 1.87 | 90.74 | 1.86 |

## D   Relation between backdoor sensitivity and the proposed pruning indices

In section Section 3.3, we define the sensitivity of neurons to backdoor by Definition 2. It is crucial to ensure that the proposed pruning indices well separate sensitive neuron with other neurons. We make scatter plots of the backdoor sensitivity v.s. differential entropy and KL divergence of neurons in some typical layers, as shown in Figure 7. The results indicate that both indices correctly pick out the most sensitive neurons, but there are more or less false positive, which may hurt benign accuracy.

## E   Robustness against out-of-distribution data

Both EP and BNP require rely on pre-activation distribution to recognize sensitive neurons, so it's natural to consider their robustness against out-of-distribution data. We use several data augmentation tricks to create distorted, low-quality data on CIFAR-10, including color jitter with 90-degree rotation (CJ) from torchvision, CoarseSaltandPepper (SAP), PolarWarping (PW), Snowflakes (Snow) from [21]. The parameters of CJ are set to be [0.5, 0.5, 0.5, 0.5]; 30% of the pixels in SAP are replaced by salt/pepper noise mask which has 1% to 10% the size of the input image; the translation percent in PW is set to be ±0.2; the snow size of Snow is (0.2, 0.5) and its speed is (0.01, 0.05). Samples of perturbed images are shown in Appendix E. Now we replace 10% of the data with out-of-distribution data, and check the defense performance of EP and BNP. The results are shown in Table 6, which illustrate that both EP and BNP remain high performance against out-of-distribution data.

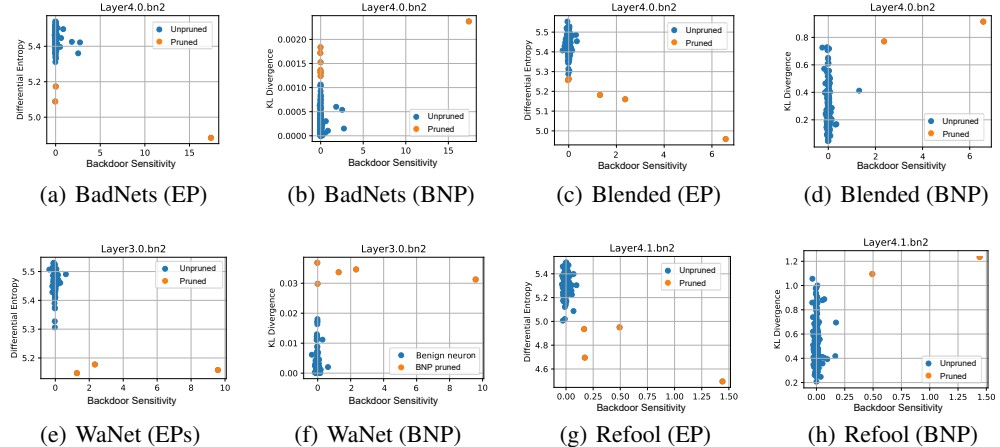

(a) BadNets (EP)  (b) BadNets (BNP)  (c) Blended (EP)  (d) Blended (BNP)

(e) WaNet (EPs)  (f) WaNet (BNP)  (g) Refool (EP)  (h) Refool (BNP)

Figure 7: Scatter plots of backdoor sensitivity of mid-layer neurons to backdoor and the corresponding differential entropy and KL divergence indices.

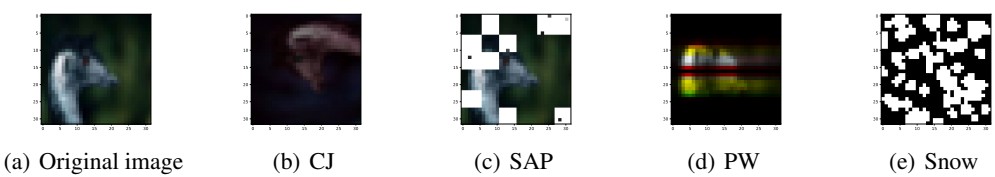

(a) Original image  (b) CJ  (c) SAP  (d) PW  (e) Snow

Figure 8: Examples of the perturbed data.

Table 6: Experimental Results of the proposed methods using low-quality datasets.

| Attacks | Origin ACC | Origin ASR | CJ ACC | CJ ASR | SAP ACC | SAP ASR | PW ACC | PW ASR | Snow ACC | Snow ASR |
|---|---|---|---|---|---|---|---|---|---|---|
| EP | 93.88 | 0.86 | 93.62 | 0.88 | 93.19 | 0.86 | 93.65 | 0.82 | 93.23 | 0.93 |
| BNP | 93.60 | 1.60 | 93.56 | 1.52 | 91.17 | 1.58 | 92.62 | 0.96 | 90.99 | 0.95 |

## F  Neuron pruning accuracy

To evaluate the accuracy in locating malicious neurons of the proposed methods and the SOTA pruning-based method ANP, we record the minimum pruning amount that can lead to acceptable degradation on ASR. Specifically, we gradually change the hyperparameter u of EP, BNP and the pruning threshold of ANP until their ASRs drop below 10%, which we regard as a sign of successful defense. The detailed results are shown in Table 7.

Under most situations, the proposed EP and BNP show higher precision in finding malicious neurons than ANP, as they achieve similar results to ANP by pruning much fewer neurons.

## G  Potential adaptive attacks

In this section, we explore some potential adaptive attacks against the proposed methods.

### G.1  BN re-parameterization against BNP

BNP rely on the correct information from the BN statistics. However, this information can be modified by a defense-aware attacker using re-parameterization techniques. Specifically, consider a

Table 7: The number of pruned neurons for EP, BNP and ANP to be able to achieve considerable defending performance.

| | EP | | | BNP | | | ANP | | |
| Attacks | ACC | ASR | #Pruned | ACC | ASR | #Pruned | ACC | ASR | #Pruned |
|---|---|---|---|---|---|---|---|---|---|
| BadNets(A2O) | 93.86 | 5.22 | 2 | 93.84 | 1.84 | 3 | 93.6 | 5.23 | 4 |
| BadNets(A2A) | 94.73 | 2.00 | 7 | 94.64 | 7.70 | 10 | 80.00 | 3.44 | 235 |
| Blended | 94.21 | 3.40 | 2 | 94.21 | 3.40 | 2 | 93.84 | 5.21 | 27 |
| CLA | 94.86 | 7.11 | 19 | 94.07 | 9.67 | 71 | 91.45 | 7.20 | 256 |
| Refool | 93.18 | 8.94 | 79 | 93.97 | 7.12 | 6 | 63.91 | 1.29 | 186 |
| IAB | 93.67 | 8.07 | 10 | 93.68 | 7.63 | 6 | 93.27 | 4.83 | 36 |
| WaNet | 93.43 | 4.06 | 43 | 94.87 | 9.64 | 13 | 94.99 | 1.31 | 3 |
| SSBA(TinyImageNet) | 66.41 | 4.13 | 5 | 66.36 | 6.04 | 5 | 66.49 | 2.81 | 7 |

BN operation performed on each neuron by:

$$\mathrm{B\tilde{N}}^{(l)}(\boldsymbol{x}_k^{(l)})_{\tilde{\gamma}_k^{(l)}, \tilde{\beta}_k^{(l)}} = \tilde{\gamma}_k^{(l)} \frac{\boldsymbol{x}_k^{(l)} - \tilde{\mu}_k^{(l)}}{\tilde{\sigma}_k^{(l)}} + \tilde{\beta}_k^{(l)}$$

$$= \frac{\tilde{\gamma}_k^{(l)}}{\tilde{\sigma}_k^{(l)}} \boldsymbol{x}_k^{(l)} + (\tilde{\beta}_k^{(l)} - \frac{\tilde{\mu}_k^{(l)}}{\tilde{\sigma}_k^{(l)}}).$$

The attacker can calculate the sample statistics denoting $\mu_k^{(l)}$ and $\sigma_k^{(l)}$ from benign data, and assign new weights and bias by:

$$\gamma_k^{(l)} = \frac{\sigma_k^{(l)}}{\tilde{\sigma}_k^{(l)}} \tilde{\gamma}_k^{(l)}$$

$$\beta_k^{(l)} = \tilde{\beta}_k^{(l)} - \frac{\tilde{\mu}_k^{(l)}}{\tilde{\sigma}_k^{(l)}} + \frac{\mu_k^{(l)}}{\sigma_k^{(l)}}$$

In this way, the whole linear transformation remains unchanged, but the BN statistics become consistent with that of the benign data, which makes the backdoor neurons unable to be detected by the proposed BNP.

Table 8: Experimental results on regularization-based adaptive attack.

| | $\lambda = 1$ | | $\lambda = 0.1$ | | $\lambda = 0.01$ | | $\lambda = 0.001$ | |
| Attacks | ACC | ASR | ACC | ASR | ACC | ASR | ACC | ASR |
|---|---|---|---|---|---|---|---|---|
| Origin | NaN | NaN | NaN | NaN | 95.11 | 100.00 | 93.41 | 75.30 |
| EP | — | — | — | — | 93.63 | 9.23 | 93.35 | 0.63 |
| BNP | — | — | — | — | 94.15 | 0.32 | 93.82 | 0.24 |

### G.2 Regularization-based Adaptive Attacks

Both of the two methods rely on the discrepancy of the benign distribution and poisoned distribution. Hence, we wonder whether EP and BNP still work if the attacker try to regularize the distribution to minimize the discrepancy. Hence, we use BadNets as an example to perform the adaptive attack with an additional objective:

$$\mathcal{L}_{adaptive} = \mathbb{E}_{(\boldsymbol{x},y)\sim\mathcal{D}}[\mathrm{D}_{\mathrm{CE}}(y, f(\delta(\boldsymbol{x}))] + \lambda \sum_{1 \le k \le K} \sum_{1 \le l \le L} \mathbb{E}_{\boldsymbol{x}\sim\mathcal{X}}[(\phi_k^{(l)} - \hat{\phi}_k^{(l)})^2], \qquad (4)$$

where $\phi_k^{(l)}$ and $\hat{\phi}_k^{(l)}$ denote the pre-activations of benign and poisoned samples in the $k^{th}$ neuron of the $l^{th}$ layer. The goal of the second term is to minimize the discrepancy between the benign distribution and the poisoned distribution with a trade-off hyperparameter $\lambda$. We select $\lambda$ from [1.000, 0.100, 0.010, 0.001] and train four models and do pruning using EP and BNP, the results are shown in Table 8. We find that too large $\lambda$ makes the objective not trainable, hence collapse the training

Table 9: Experimental results of our methods against some existing adaptive attacks.

| | ML-MMD | | WB | |
|---|---|---|---|---|
| | ACC | ASR | ACC | ASR |
| Origin | 87.51 | 98.48 | 92.12 | 100 |
| EP | 86.34 | 3.26 | 90.88 | 24.65 |
| BNP | 86.9 | 36.15 | 90.88 | 37.4 |

process. This may because it is not realistic to distinguish benign and poisoned samples while making their distribution indistinguishable. Decreasing $\lambda$ to below 0.01 makes the model trainable. Even in this case, we find EP and BNP are still able to remove the backdoor without influence too much on the clean accuracy.

A recently proposed method[44] also regularizes the distribution discrepancy between benign samples and poisoned samples by Multi-Level Maximum Mean Discrepancy (ML-MMD), which is a direct attack against our defense. Besides, [11] propose to use Wasserstein distance to minimize the difference between poison and clean data latent feature distribution (WB). We test our methods on them and find they still work with some degradation. The results are shown in Table 9. The main reason for such degradation is the strong constraint added to minimize the difference between clean and poison data latent feature distribution. Besides, both the above attacks adds constraint to loss function, making it harder to converge during training process, and their final ACC is obviously lower than normal attacks.

## H   Ablation results on hyperparameter

In this section, we provide an ablation study on the choice of the threshold hyperparameter $u_h/u_k$ against more attacks. As shown in Figure 9, both EP and BNP gives promising defending results against most tested attacks on CIFAR 10 when $u_h/u_k$ is around 3. Besides, the robustness of the hyperparameter is supported by the wide choice of available $u_h/u_k$ that lead to acceptable results.

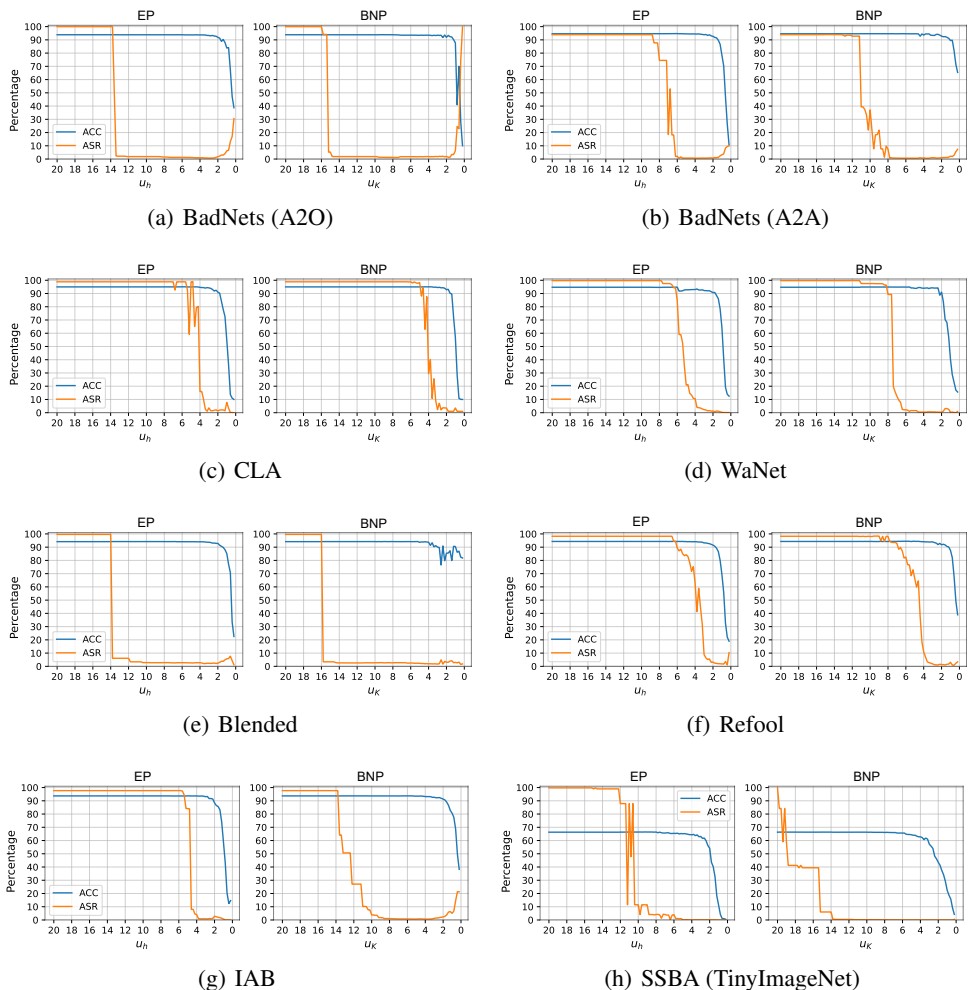

Figure 9: The performance of proposed EP and BNP with different hyperparameter $u_h/u_k$ against all the tested attacks on ResNet-18.