# OpenReview forum: "Pre-activation Distributions Expose Backdoor Neurons"
_NeurIPS.cc/2022/Conference — NeurIPS 2022 Accept_

### Official Review · Reviewer_Ton2 · 2022-07-05

**Rating:** 6
**Confidence:** 4
**Soundness:** 3 good
**Presentation:** 3 good
**Contribution:** 2 fair

**Summary:**

This paper reveals that poisoned neurons and benign neurons have different behaviors, measured by the entropy of neuron activations and the BN statistics. Motivated by these understandings, the authors propose DDE and MBNS targeting the defense of poison-only backdoor attacks and attacked pre-trained models, respectively. The proposed methods are evaluated on both CIFAR-10 and Tiny-ImageNet datasets under seven attacks.

**Questions:**


Cons
1. Some statements need further support or modifications to avoid over-claim.
- ‘we claim that both the proposed indices can perfectly separate...’ (Line 60-62, Page 2): This statement should be modified. This claim can be made only if your methods have solid theoretical foundations.
- ‘there was no such quantity defined in the literature to measure...’ (Line 139-140, Page 4): This is an over-claim. Specifically, all existing pruning-based defenses provided a quantity (e.g., activation values on benign samples) to measure it.
2. Missing some important related works.
- No discussion about pre-processing-based backdoor defenses (e.g., [1-3])
- Missing two poison-suppression-based backdoor defenses [4, 5].
3. My biggest concern is about the experiment parts.
- How to select the hyper-parameter u? Do you use the same u under different attacks on all datasets?
- To ensure a fair comparison among all pruning-based defenses, I would like to see the results (i.e., ACC and ASR) with respect to the pruning ratio. It is necessary to verify whether the proposed methods are truly better in finding malicious neurons.
- I am interested in whether the proposed defenses are truly resistant to adaptive methods since only poison-only backdoor attacks are used in the evaluations. The author should also test their methods on some training-controlled attacks (e.g., [6, 7]), which aim to reduce the difference between poisoned samples and benign samples in the hidden feature space.


Minor Comments
1. The ‘we will’ should be ‘we’ in Line 36 (Page 1).
2. There are two typos in Table 2: ‘0.1’-->‘0.10’, ‘64.2’-->‘64.20’.
3. Please recheck and cite the official version of all references (e.g., [8-9]).


**Limitations:**

The authors fail to adequately mention the limitations and potential negative societal impact of their work. Please discuss them explictly.

**Strengths And Weaknesses:**

Pros
1. The topic is of sufficient significance and interest to NeurIPS audiences.
2. The authors discuss the resistance to potential adaptive attacks, which should be encouraged.
3. In general, the idea is easy to follow.
4. The authors discuss the efficiency of different defenses, which should be encouraged.


In general, I enjoy reading this paper. However, I still have some concerns about this paper. I will increase my scores if the authors can (partly) address my concerns. The detailed comments are as follows:


Cons
1. Some statements need further support or modifications to avoid over-claim.
- ‘we claim that both the proposed indices can perfectly separate...’ (Line 60-62, Page 2): This statement should be modified. This claim can be made only if your methods have solid theoretical foundations.
- ‘there was no such quantity defined in the literature to measure...’ (Line 139-140, Page 4): This is an over-claim. Specifically, all existing pruning-based defenses provided a quantity (e.g., activation values on benign samples) to measure it.
2. Missing some important related works.
- No discussion about pre-processing-based backdoor defenses (e.g., [1-3])
- Missing two poison-suppression-based backdoor defenses [4, 5].
3. My biggest concern is about the experiment parts.
- How to select the hyper-parameter u? Do you use the same u under different attacks on all datasets?
- To ensure a fair comparison among all pruning-based defenses, I would like to see the results (i.e., ACC and ASR) with respect to the pruning ratio. It is necessary to verify whether the proposed methods are truly better in finding malicious neurons.
- I am interested in whether the proposed defenses are truly resistant to adaptive methods since only poison-only backdoor attacks are used in the evaluations. The author should also test their methods on some training-controlled attacks (e.g., [6, 7]), which aim to reduce the difference between poisoned samples and benign samples in the hidden feature space.


Minor Comments
1. The ‘we will’ should be ‘we’ in Line 36 (Page 1).
2. There are two typos in Table 2: ‘0.1’-->‘0.10’, ‘64.2’-->‘64.20’.
3. Please recheck and cite the official version of all references (e.g., [8-9]).


References
1. DeepSweep: An Evaluation Framework for Mitigating DNN Backdoor Attacks using Data Augmentation. AsiaCCS, 2021.
2. Backdoor Attack in the Physical World. ICLR Workshop, 2021.
3. Neural Trojans. ICCD, 2017.
4. Backdoor Defense via Decoupling the Training Process. ICLR, 2022.
5. Anti-Backdoor Learning: Training Clean Models on Poisoned Data. NeurIPS, 2021.
6. Backdoor Attack with Imperceptible Input and Latent Modification. NeurIPS, 2021.
7. Bypassing Backdoor Detection Algorithms in Deep Learning. EuroS&P, 2020.
8. Backdoor Learning: A Survey. IEEE TNNLS, 2022.
9. Backdoor Scanning for Deep Neural Networks through K-Arm Optimization. ICML, 2021.

---

> ### Author Response · Authors · 2022-08-02
> **Reply to Reviewer Ton2**
>
> We thank you the reviewer for the delicate reading and valuable comments. The following is the response to the reviewer:
>
> Questions:
>
> 1. We agree with the reviewer that these statements are overclaimed. We will carefully check the statements in the manuscript again.
>
> 2. Thanks for pointing out the important related works that we've missed. We will discuss them in the final version.
>
> 3. Experiments
>
> 3.1 We use 3 as the threshold hyperparameters u on CIFAR-10 and 4 on Tiny-ImageNet for both DDE and MBNS against all the tested attacks. Choosing an optimal threshold can lead to even better performance. It's our negligence that we do not clarify this well in the manuscript. To better demonstrate the robustness of the choice of hyperparameters, we will provide a figure showing how the choice of hyperparameters affects the defense performance under different attacks in Appendix.G - Figure 9 in the revision version of the manuscript, and more detailed comments will be added in the final version.
>
> 3.2 To evaluate the accuracy in locating malicious neurons of the proposed methods and the SOTA pruning-based method ANP, we record the minimum pruning amount that can lead to acceptable degradation on ASR. Specifically, we gradually change the hyperparameter u of DDE, MBNS and the pruning threshold of ANP until their ASRs drop below 10%, which we regard as a sign of successful defense. The detailed results are shown below:
>
> |                    |   | DDE   |      |         |   | MBNS  |      |         |   | DDE   |      |         |
> |--------------------|---|-------|------|---------|---|-------|------|---------|---|-------|------|---------|
> |                    |   | ACC   | ASR  | #Pruned |   | ACC   | ASR  | #Pruned |   | ACC   | ASR  | #Pruned |
> | BadNets(A2O)       |   | 93.86 | 5.22 | 2       |   | 93.84 | 1.84 | 3       |   | 93.60 | 5.23 | 4       |
> | BadNets(A2A)       |   | 94.73 | 2.00 | 7       |   | 94.64 | 7.70 | 10      |   | 80.00 | 3.44 | 235     |
> | Blended            |   | 94.21 | 3.40 | 2       |   | 94.21 | 3.40 | 2       |   | 93.84 | 5.21 | 27      |
> | CLA                |   | 94.86 | 7.11 | 19      |   | 94.07 | 9.67 | 71      |   | 91.45 | 7.20 | 256     |
> | Refool             |   | 93.18 | 8.94 | 79      |   | 93.97 | 7.12 | 6       |   | 63.91 | 1.29 | 186     |
> | IAB                |   | 93.67 | 8.07 | 10      |   | 93.68 | 7.63 | 6       |   | 93.27 | 4.83 | 36      |
> | WaNet              |   | 93.43 | 4.06 | 43      |   | 94.87 | 9.64 | 13      |   | 94.99 | 1.31 | 3       |
> | SSBA(TinyImageNet) |   | 66.41 | 4.13 | 5       |   | 66.36 | 6.04 | 5       |   | 66.49 | 2.81 | 7       |
>
> Under most tested attacks, DDE and MBNS need less neuron pruning than ANP, indicating that they are more accurate in detecting malicious neurons. Besides, the tuning of ANP requires significantly more time and effort due to its intensive mask values.
>
>
> 3.3 We have tested our methods against Multi-leverl Maximum Mean Discrepancy (ML-MMD) enhanced backdoor attack [1], which is a training-controlled attack in section E.1.2 of our appendix. Experimental results show that our pruning methods successfully mitigate the backdoored model. In addition, we conduct extra experiments to show our defending performance against Wasserstein Backdoor attack (WB) [2]. The results are shown below:
>
> |        |   | ML-MMD |       |   | WB    |        |
> |--------|---|--------|-------|---|-------|--------|
> |        |   | ACC    | ASR   |   | ACC   | ASR    |
> | Origin |   | 87.51  | 98.48 |   | 92.12 | 100.00 |
> | DDE    |   | 86.34  | 3.26  |   | 90.88 | 24.65  |
> | MBNS   |   | 86.93  | 6.15  |   | 90.88 | 37.40  |
>
> There is some degradation on the performance of our methods, mainly due to the strong constraint added to minimize the difference between clean and poison data latent feature distribution. Besides, both the above attacks adds constraint to loss function, making it harder to converge during training process, and their final clean acc is obviously lower than normal attacks.
>
> [1] Xia P, Niu H, Li Z, et al. Enhancing Backdoor Attacks with Multi-Level MMD Regularization[J]. IEEE Transactions on Dependable and Secure Computing, 2022.
> [2] Doan K, Lao Y, Li P. Backdoor attack with imperceptible input and latent modification[J]. Advances in Neural Information Processing Systems, 2021, 34: 18944-18957.

---

> > ### Comment · Reviewer_Ton2 · 2022-08-03
> > **Post-rebuttal Comments**
> >
> > I would like to thank the authors for their detailed response. I think the authors have addressed most of my concerns. Besides, I believe that the analysis of BN statistics is an important but currently neglected defense method. As such, I increase my score to 'Weak Accept'.
> >
> > Just one more warm notification: please don't forget to deal with the 'Minor Comments' :)

---

> > > ### Author Response · Authors · 2022-08-03
> > > **Reply to Post-rebuttal Comments of Reviewer Ton2**
> > >
> > > Thank you for the appreciation of our work. We will check our paper through carefully and include this valuable discussion in the final version of our paper to make it a more solid work. Thanks again for your detailed reading and rectification.

---

### Official Review · Reviewer_WDQn · 2022-07-09

**Rating:** 4
**Confidence:** 5
**Soundness:** 3 good
**Presentation:** 3 good
**Contribution:** 2 fair

**Summary:**

In this paper, the authors propose a defense approach to mitigate the backdoor attacks on convolution neural networks (CNNs). Specifically, the authors first observe that the neuron activation distributions within the backdoor-infected CNN are different for benign and backdoored samples. Based on such an observation, the authors further propose to calculate the differential entropy to distinguish the compromised neuron. In the experiment section, the authors evaluate the proposed approach to CIFAR-10 and TinyImageNet benchmarks under different settings(e.g., training phase, post-training phase, etc). Moreover, the authors also test the robustness of the proposed approach on two adaptive attacks. The results demonstrate that the proposed method can significantly outperform comparison work in the dimensions of robustness, efficacy, and efficiency.


**Questions:**

1. Is this work can still perform efficacy under multiple-label and other practical scenarios?


2. What device for you to test the running time in the experiment section?

**Limitations:**

Requiring clean validation data makes the approach somehow impractical.

**Strengths And Weaknesses:**

+++++++Strengths+++++++

1. Well-structured presentation and easy to follow.
2. Comprehensive evaluation.
3. Considering two adaptive attacks.

-------------Weakness------------

1. The novelty is somehow limited. Since inspecting compromised neurons for defending against backdoor attacks has been widely adopted by previous work, this work may not seem that novel.

2.  I think more previous unlearning-based work or training-phase stage defense should also be considered for comparison. For example, DBD[1] and the most recent approach[2].

3. Missing evaluation on several practical scenarios. I think the proposed approach should also be evaluated under some rather practical scenarios, for example,  multiple-label scenarios, and multiple backdoors within the same infected label which previous work (neural cleanse) has discussed that.

[1]https://openreview.net/forum?id=TySnJ-0RdKI
[2]ADVERSARIAL UNLEARNING OF BACKDOORS VIA IMPLICIT HYPERGRADIENT

---

> ### Author Response · Authors · 2022-08-02
> **Reply to Reviewer WDQn**
>
>
> Weakness:
>
> 1. It is right that inspecting compromised neurons for defending backdoor attacks have been studied in some works. However, different methods identify compromised neurons according to different principles, and the principles used for inspection are the main contribution of these studies that we should focus on, instead of inspection for defense itself. As far as we know, our idea that leverages the feature discrepancy and BN statistics to identify compromised neurons has not been explored before.
>
> 2. We thank the reviewer for the insightful suggestion. Indeed, the setting of training-phase defense is consistent with that of DDE. Hence, it is reasonable to include training-phase defense in the discussion of this paper. We will reproduce them on our benchmark for fair comparison in the final version.
>
> 3. To address the reviewer's concern, we conduct additional experiments on multiple-backdoor settings following Neural Cleanse (Multiple Infected Labels with Separate Badnets Triggers). The results are shown below (four backdoors):
>
> | Origin |       |       |       |       |   | DDE   |       |       |       |       |   | MBNS  |       |       |       |       |
> |--------|-------|-------|-------|-------|---|-------|-------|-------|-------|-------|---|-------|-------|-------|-------|-------|
> | ACC    | ASR-0 | ASR-1 | ASR-2 | ASR-3 |   | ACC   | ASR-0 | ASR-1 | ASR-2 | ASR-3 |   | ACC   | ASR-0 | ASR-1 | ASR-2 | ASR-3 |
> | 93.40  | 98.11 | 98.34 | 98.23 | 94.19 |   | 93.07 | 1.93  | 0.29  | 0.91  | 4.91  |   | 93.10 | 1.48  | 0.26  | 0.81  | 3.49  |
>
> The preliminary results show that the performance of the proposed method is hardly affected by the number of injection backdoors.
>
> Questions:
>
> 1. The preliminary results show that the performance of the proposed method is hardly affected by the number of injection backdoors, please see our response to Weakness 3 for the detailed results.
>
> 2. All the methods are tested on a single NVIDIA RTX 2080Ti GPU. We omit the time of hyperparameters tuning and evaluation, and directly record the training/pruning time.
>
> Limitations:
>
> 1. We thank the reviewer's suggestions, but we respectfully disagree with the reviewer for the following reasons:
>
> (1) We propose two methods (DDE and MBNS) that can be used under two different defense settings. Please refer to section 4.4 for the details about the settings. In short, MBNS need clean holdout data, but DDE does not. DDE is used when only the original training (poisoned) dataset is given.
>
> (2) Most of the previous studies also require clean data as well as the labels to repair the infected networks (FP, NAD, ANP, etc.). This is widely acknowledged as a common backdoor defense scenario. Besides, compared with the above-mentioned methods, MBNS can work on clean data _without_ label information. It only needs to collect the BN statistics and does not require labels to be involved into calculation. Such property further alleviates the requirements of MBNS on clean data.
>
> (3) The two methods (DDE, MBNS) are proposed to defend backdoor attacks under two different defense settings (two out of three common settings, please refer to TABLE I in [1]) based on one principle (discrepancy on feature distribution). To the best of our knowledge, almost all of the previous studies focus on addressing only one of these settings. We believe that the generality of our proposed principle is one of the advantages rather than a limitation and makes it possible for users to choose the better way under specific settings.
>
> [1] Li Y, Jiang Y, Li Z, et al. Backdoor learning: A survey[J]. IEEE Transactions on Neural Networks and Learning Systems, 2022.

---

> > ### Comment · Reviewer_WDQn · 2022-08-08
> > **Response to the rebuttals**
> >
> > Indeed, I agree with you that almost all of the previous studies focus on addressing only one of these settings. However, you should also compare the state-of-art approach for each setting. This is because defenders may have an incentive to implement them simultaneously for defense purposes. Also, comparing with the state-of-art approach for each setting is also better for the readers/reviewers to understand the potential and efficacy of your approach.  Moreover, a comparison to the mixed version(mix defense approach for the two settings) should also be considered.

---

> > > ### Author Response · Authors · 2022-08-09
> > > **Reply to Post-rebuttal Response of Reviewer WDQn**
> > >
> > > __"However, you should also compare the state-of-art approach for each setting. This is because defenders may have an incentive to implement them simultaneously for defense purposes. Also, comparing with the state-of-art approach for each setting is also better for the readers/reviewers to understand the potential and efficacy of your approach."__
> > >
> > > We agree with the reviewer that we should compare SOTA approaches of each setting. As we already compare our methods with those require benign holdout data in the paper, we additionally give the results of two other SOTA methods (DBD [1], ABL [2]) which require poisoned dataset as DDE does. The results are shown below:
> > >
> > > |            |   | BadNets(A2O) |          |   | BadNets(A2A) |          |   | CLA       |          |   | Blended   |          |   | Refool    |          |   | IAB       |          |   | WaNet     |          |
> > > |------------|---|--------------|----------|---|--------------|----------|---|-----------|----------|---|-----------|----------|---|-----------|----------|---|-----------|----------|---|-----------|----------|
> > > |            |   | ACC          | ASR      |   | ACC          | ASR      |   | ACC       | ASR      |   | ACC       | ASR      |   | ACC       | ASR      |   | ACC       | ASR      |   | ACC       | ASR      |
> > > | ABL        |   | 90.28        | __0.18__ |   | 65.87        | 21.59    |   | 70.72     | 32.40    |   | 78.16     | __0.19__ |   | 87.64     | __0.04__ |   | 68.98     | 97.70    |   | 88.67     | __2.82__ |
> > > | DBD        |   | 90.57        | 1.86     |   | 93.75        | 0.78     |   | 90.76     | 95.31    |   | 89.56     | 28.76    |   | 92.19     | 11.72    |   | 92.59     | __0.04__ |   | 88.28     | 4.18     |
> > > | DDE (Ours) |   | __93.88__    | 0.86     |   | __94.49__    | __0.61__ |   | __94.42__ | __0.91__ |   | __93.67__ | 2.24     |   | __93.35__ | 8.90     |   | __93.17__ | 0.94     |   | __93.35__ | 8.90     |
> > >
> > > The advantage of training-phase defense is that it is able to reduce ASR closer to 0, but it also downgrades ACC largely. If we are not demanding extremely low ASR, then our pruning-based approach offers a much better trade-off between ACC and ASR. It is worth noting that DBD, which performs better than ABL, involves self-supervised pretraining and is considered time-consuming. In terms of more details, DBD runs for 8 hours on a NVIDIA 2080Ti GPU, while our methods finish the pruning in several seconds, and require less than 2 hours even if we take the training time into consideration.
> > >
> > > We will report more results in the final version.
> > >
> > > __"Moreover, a comparison to the mixed version(mix defense approach for the two settings) should also be considered."__
> > >
> > > We respectfully disagree with the reviewer for the following reasons:
> > >
> > > 1. A mixed version of defense require both poisoned dataset and benign dataset, which is a far more strict requirement. As far as we know, almost all the mainstream methods do not consider such a defense setting in their studies.
> > >
> > > 2. We kindly remind you that DDE and MBNS are two separate methods for different settings. Each of them only require one of the datasets (either a clean dataset or a poisoned dataset). It is definitely unfair to compare our methods with a mixed version of defense, which require two different datasets.
> > >
> > > 3. It is still unclear whether mixing different defense method will improve the defense effect. For example, using two defense method sequentially may further degrade the clean accuracy and make the loss outweigh the gain. Moreover, if we want to mix defense of, for example, setting A and setting B, there will be $n \times m$ mixing strategies if we have $n$ methods for setting A and $m$ methods for setting B. We argue that it requires a thoughtful discussion and systematical study on what to mix and how to mix, instead of just posting the results in the paper. We believe that this can be a future work, but is far beyond the scope of this paper.
> > >
> > >
> > > [1] Huang, K., Li, Y., Wu, B., Qin, Z. and Ren, K., 2021, September. Backdoor Defense via Decoupling the Training Process. In International Conference on Learning Representation, 2022.
> > >
> > > [2] Li, Y., Lyu, X., Koren, N., Lyu, L., Li, B. and Ma, X., 2021. Anti-backdoor learning: Training clean models on poisoned data. Advances in Neural Information Processing Systems, 2021.

---

> ### Author Response · Authors · 2022-08-07
> **Follow-up discussion**
>
> Dear Reviewer WDQn, thanks for your insightful suggestions. We've tried to address your concerns, please let us know if you have further comments. We're looking forward to your reply.

---

### Official Review · Reviewer_Yu18 · 2022-07-10

**Rating:** 8
**Confidence:** 3
**Soundness:** 3 good
**Presentation:** 3 good
**Contribution:** 4 excellent

**Summary:**

In this paper, the authors studies how to better defend models from backdoor attacks. Specifically, they identified that there are only a subset of neurons, which are responsible for the poisoned behaviors. The found that if one prunes these poisoned neurons, then the backdoor behavior can be effectively stopped. The authors found a simple way of identifying such neurons through the two measurements: discrepancy of differential entropy and mismatched batchnorm statistics. By using these two measurements, they are able to more effectively stop the backdoor behavior while requiring very few additional compute. The also can identified such backdoor neurons with as few as 10 benign examples.





**Questions:**

I don't have any questions about the paper as of now.

**Limitations:**

The authors have adequately address the limitation of their approach and societal impact of their work.

**Strengths And Weaknesses:**

**Strength**
- The biggest strength of the paper is the method's effective as well as simplicity
- The presentation of the paper is very clear and easy to follow
- The experiments are thorough and convincingly shows the benefit of the proposed approach.

**Weakness**
- I do not find any obvious weakness in the paper. The wording may be improved slightly, but it doesn't impact the message of the paper very much.
- Given its superior effectiveness in defending against the standard backdoor attacks in standard settings, I am curious whether it can also help stop the backdoor attacks in self-supervised or unsupervised settings. I think the contribution of the paper is already solid, but am simply interested in learning more about its implication for other backdoor settings.

Overall, I really enjoy reading the paper, and recommend the paper being accepted.

---

> ### Author Response · Authors · 2022-08-02
> **Reply to Reviewer Yu18**
>
> Weakness:
>
> 1. We thank the reviewer for the appreciation of our work. We will carefully polish the writing of the manuscript.
>
> 2. It is interesting to discuss about the defense performance when attacks happened in self-supervised/unsupervised settings. Our methods are proposed based on the observation that the feature distribution of benign data and poisoned data are different in supervised setting. However, this property may not hold in self-supervised/unsupervised settings. Hence, we cannot claim that our method will work on such settings. Due to the time limit, we will consider adding discussion about these settings in the final version of the paper.

---

### Official Review · Reviewer_Bhnv · 2022-07-10

**Rating:** 5
**Confidence:** 3
**Soundness:** 2 fair
**Presentation:** 3 good
**Contribution:** 3 good

**Summary:**

Based on the observation that clean and backdoor training samples have different feature statistics, the paper proposes to detect and remove the neurons that are mostly affected by backdoor training samples. Empirical results show the effectiveness of the proposed method under certain scenarios.

**Questions:**

The title in the submitted PDF is different from that in Openreview.

**Limitations:**

Please see above.

**Strengths And Weaknesses:**

Strength:
1. The proposed method is intuitively correct: Clean samples and backdoor samples should have somehow different feature statistics. This is because the model overfits superficial correlation on backdoor samples while it learns semantic information on clean samples. I don't doubt the proposed method is effective in improving backdoor robustness.
2. The paper is well-written and easy to follow. All notations and terms are defined clearly without ambiguous.
3. The proposed method can defend some backdoor attacks when few clean samples are available.

Weakness:
1. My main concern is how such good performance reported in this paper can be generalized to different settings.

1.1 In practice, we don't know what type of backdoor attacks are used. So, it is very important for the defense method to be generally robust against multiple different attacks using the **same** hyper-parameters in the defense method. However, in this paper, the authors only mentioned the only hyperparameter $\mu_k$ "is **usually** set to 3". The use of the word "usually" makes me feel skeptical about the results. The readers need to know what value $\mu_k$ is used under all cases (on all datasets and against all attacks). The ideal case is we have a value that works well for all attacks, even under the mixture attack setting (i.e. when multiple different attack simultaneous exists in the training set). But that should be challenging in my point of view since the distributions of backdoor samples from different attacks can be very different. So what are the "unusual" cases in this paper? If different hyper-parameter values are used against different attacks in the proposed method, then it is unfair comparison with the baselines. As far as I know, NAD can achieve better performance if the hyper-parameters are tuned for different attacks.
Again, I don't doubt for a second that the proposed method can help defense backdoor attacks. I just concern whether it can achieve such good performance across all reported different attacks using the same hyper-parameter setting.
It won't harm the contribution of this paper if the performance drops a little bit when the same $\mu_k$ value is used for different attacks.

1.2 Please consider more different attack settings. For example, when a different poisoning ratio (eg 5%, 20%, 30%...) is used? Does the proposed method need to use a different uk value under different poisoning ratios to achieve good performance?

1.3 Please provide ablation study results on different uk values.


2. In Figure 5 (c) in appendix, the visualization of Blend attack looks a bit weird to me. I can't see any effect of the blend image (which is usually a hello kitty image or strong random noise). Please see Figure 5 in [1] or the original Blend attack paper for example. May I ask what blending ratio and blend image/pattern you are using?

I will improve my score if the concerns are addressed.

---

> ### Author Response · Authors · 2022-08-02
> **Reply to Reviewer Bhnv**
>
> We thank the reviewer for the insightful comments. We address the concerns below:
>
> Weakness:
>
> 1.1. We use 3 as the threshold hyperparameters u on CIFAR-10 and 4 on Tiny-ImageNet for both DDE and MBNS against all the tested attacks. Choosing an optimal threshold can lead to even better performance. It's our negligence that we do not clarify this well in the manuscript. To better demonstrate the robustness of the choice of hyperparameters, we will provide a figure showing how the choice of hyperparameters affects the defense performance under different attacks in Appendix.G - Figure 9 in the revision version of the manuscript, and more detailed comments will be added in the final version.
>
> 1.2. We have evaluated our methods when the poisoning rate is 1%, 5% and 10%, please refer to Table 4 in the Appendix for detailed results. For higher poisoning rates, i.e., 20% and 30%, we add the following experiments tested on BadNets due to the time limit. The results are shown below:
>
> |     |   | Origin |       |   | DDE   |      |   | MBNS  |      |
> |-----|---|--------|-------|---|-------|------|---|-------|------|
> |     |   | ACC    | ASR   |   | ACC   | ASR  |   | ACC   | ASR  |
> | 20% |   | 93.23  | 99.46 |   | 92.22 | 0.66 |   | 91.10 | 1.84 |
> | 30% |   | 92.47  | 99.62 |   | 92.59 | 1.32 |   | 92.31 | 1.49 |
>
> The drop in clean accuracy is the least when the poisoning rate is 30%. We hypothesize that the bimodal phenomenon is more significant when the amount of benign data and poisoned data are closer (i.e., the poisoning rate is closer to 50%). In this case, the pruning based on distribution discrepancy should be more accurate. A similar trend can be observed in Table 4 that attacks with smaller poisoning rate are mostly harder to defense. Note that we use u=3 for all the experiments on CIFAR-10, regardless of the poisoning ratios.
>
> 1.3. We provide the results in the revised verion. Please refer to Appendix.G - Figure 9 for the detailed results.
>
> 2. We generate a 3x3 random gaussian noise as the trigger pattern. Please check it from the lower right corner of the image. The blending ratio is set to 20%, so the pattern is not that noticeable.

---

> > ### Comment · Reviewer_Bhnv · 2022-08-03
> > **Follow up comments and questions**
> >
> > Thank you for your careful response.
> >
> > My concern on the hyper-parameter selection issue is solved. I now think more positive about this work.
> >
> > However, I don't see the updated Figure 9 for ablation study on the $\mu_k$ value.
> >
> > And I have a new question: Why do you use a 3x3 square for Blend attack? The original Blend attack paper used either the Hello Kitty image or a global random noise as the trigger. I'm not saying your modified version of Blend attack is meaningless. At least it has around 100% attack success rate on normal training method. I'm just curious why you use this unusual (and even unique as far as I know) design of Blend attack? In my opinion, the Global trigger used in Blend attack is an important property differentiating it from other backdoor attacks using local patches such as the BadNet and Clean Label attack. Is it possible to show results of Blend attack with the global pattern? It doesn't matter if your method success or fails on it, since you already have good results on other six attacks. It is only for completeness of the experiments. If it doesn't work as well as other attacks, you can still report it in appendix as a failure case. If you insist on using the current local Blend attack, I'm also fine with it. But please make it clear that your "Blend" attack is not the common one that comes into the readers' mind. Otherwise, it would be confusing.

---

> > > ### Author Response · Authors · 2022-08-03
> > > **Reply to the follow up comments**
> > >
> > > We have double checked that the attached pdf is the latest version that contains the updated figure. Please let us know if you still can't see it.
> > >
> > > It is true that there are global patterns used in the original implementation of blend attack, but there is also local pattern like sunglasses introduced in the paper. We have no objection to what you said that the global pattern is an important property of blend attack, but we used to think the core part of blend attack is the _blend_ operation instead of using a global pattern. That's why we only tested 3x3 square. We agree that it is better to follow the original setting, hence we conduct experiments using a global noise pattern instead. The results are shown below:
> > >
> > > |        |   | ACC   | ASR    |
> > > |--------|---|-------|--------|
> > > | Origin |   | 94.28 | 99.82  |
> > > | FT     |   | 92.99 | 97.93  |
> > > | FP     |   | 92.66 | 99.63  |
> > > | l_inf  |   | 89.61 | 100.00 |
> > > | NAD    |   | 92.45 | 93.16  |
> > > | ANP    |   | 91.55 | 93.86  |
> > > | DDE    |   | 92.87 | 13.36  |
> > > | MBNS   |   | 91.80 | 1.07   |
> > >
> > > It seems right that defense against blend attack with a global pattern is more challenging. We will consider using the original implementation as the benchmark in the final version. Thanks again for the valuable suggestions.

---

> > > ### Author Response · Authors · 2022-08-06
> > > **Follow up comments**
> > >
> > > Thanks again for the valuable comments. We wonder if our response has addressed your concerns. Your appreciation is important to our work.

---

### Meta-Review · Area_Chair_8XG6 · 2022-08-28

**Recommendation:** Accept
**Confidence:** Certain

**Metareview:**

The authors propose a hypothesis that backdoor neurons in an infected neural network have a mixture of two distributions with significantly different moments, formed by benign samples and poisoned samples, respectively. They then propose two mathematically informed and intuitive ways to defend against the attack. The method also seems general against most types of backdoor attacks, and the evaluations give more confidence in that direction. The evaluation is extensive and includes most of the state of the art as comparison. An additional advantage is that this method has better runtime than most.

The most critical reviewer, WDQn, was concerned with the lack of evaluations against multiple attack models (multiple label attack, etc) and  the lack of comparisons against SOTA. The authors have responded with detailed and appropriate results in their last response.

Concerns from other reviewers, such as the concern about the robustness of hyperparameter u, were alleviated through ablations/results posted in the authors' rebuttal.

I therefore recommend accept.

**Award:**

No

---

### Decision · Program_Chairs · 2022-09-14

Accept